# Relative Efficacy of Weight Management, Exercise, and Combined Treatment for Muscle Mass and Physical Sarcopenia Indices in Adults with Overweight or Obesity and Osteoarthritis: A Network Meta-Analysis of Randomized Controlled Trials

**DOI:** 10.3390/nu13061992

**Published:** 2021-06-10

**Authors:** Shu-Fen Chu, Tsan-Hon Liou, Hung-Chou Chen, Shih-Wei Huang, Chun-De Liao

**Affiliations:** 1College of Nursing and Health Management, Shanghai University of Medicine and Health Sciences, Shanghai 201318, China; zhusf@sumhs.edu.cn; 2Department of Physical Medicine and Rehabilitation, Shuang Ho Hospital, Taipei Medical University, New Taipei City 235041, Taiwan; peter_liou@s.tmu.edu.tw (T.-H.L.); 10462@s.tmu.edu.tw (H.-C.C.); 13001@s.tmu.edu.tw (S.-W.H.); 3Department of Physical Medicine and Rehabilitation, School of Medicine, College of Medicine, Taipei Medical University, Taipei 110301, Taiwan; 4Master Program in Long-Term Care, Taipei Medical University, College of Nursing, Taipei 110301, Taiwan

**Keywords:** osteoarthritis, sarcopenia, diet, exercise training, muscle mass, physical function

## Abstract

Aging and osteoarthritis are associated with high risk of muscle mass loss, which leads to physical disability; this loss can be effectively alleviated by diet (DI) and exercise (ET) interventions. This study investigated the relative effects of different types of diet, exercise, and combined treatment (DI+ET) on muscle mass and functional outcomes in individuals with obesity and lower-limb osteoarthritis. A comprehensive search of online databases was performed to identify randomized controlled trials (RCTs) examining the efficacy of DI, ET, and DI+ET in patients with obesity and lower-extremity osteoarthritis. The included RCTs were analyzed through network meta-analysis and risk-of-bias assessment. We finally included 34 RCTs with a median (range/total) Physiotherapy Evidence Database score of 6.5 (4–8/10). DI plus resistance ET, resistance ET alone, and aerobic ET alone were ranked as the most effective treatments for increasing muscle mass (standard mean difference (SMD) = 1.40), muscle strength (SMD = 1.93), and walking speed (SMD = 0.46). Our findings suggest that DI+ET is beneficial overall for muscle mass in overweight or obese adults with lower-limb osteoarthritis, especially those who are undergoing weight management.

## 1. Introduction

Osteoarthritis (OA) or degenerative arthritis is the most prevalent musculoskeletal disorder worldwide [1], with rapid increases in its prevalence after the sixth decade of lifespan and with a strong impact on the health of the aging population [2]. OA commonly affects knee and hip joints, and it impairs muscle morphology and function, leading to physical disability [3,4,5], especially in individuals with obesity [6,7]. Observational studies have indicated that older individuals with OA exhibited lower muscle mass or volume than healthy controls [3,5,8,9], and having less muscle was further associated with lower muscle strength and poor functional outcomes [10,11]. Recent studies have reported that OA was associated with high risk of sarcopenia [12,13], an age-related condition characterized by attenuated muscle mass [14,15].

Obesity and sarcopenia have become public health concerns in the older population, and sarcopenic obesity might synergistically increase the risk of physical disability [16,17]. Such compounding effects of sarcopenic obesity may affect OA [18] because obesity has become epidemic in the OA population [19] and sarcopenic obesity is closely associated with OA [20]. Because obesity exerts negative effects on physical function in obese individuals with OA [7,19,21] and a high percentage body fat (BF%) is significantly associated with sarcopenia [22], people with overweight or obesity and knee or hip OA have a high risk of physical disability due to sarcopenic obesity.

Among the nonpharmacological and nonsurgical treatments for OA, exercise therapy and weight loss (for individuals with obesity) have been recommended as first-line treatments [23]. According to well-established evidence, exercise interventions benefit muscle mass and functional outcomes [24,25], and weight loss through an energy-restricted diet improves pain and function in patients with OA [26,27,28]. In particular, successful diet-induced weight loss for older adults with obesity may reduce both lean body mass (LBM) and fat-free mass (FFM) [29,30,31], with 3.8% and 15–25% of weight loss as LBM [31] and FFM [29], respectively; in addition, nearly 5% loss of leg lean mass may occur after weight management for those with obesity and OA [31]. Accordingly, for patients with OA who are at high risk of sarcopenia, muscle mass should be retained or even increased as much as possible during a weight-loss intervention through increased protein intake (e.g., high-protein diet or protein supplementation) and exercise [32,33]. However, the optimal treatment combination remains unclear due to the variety of exercise training modes for increasing muscle mass and strength [34] and the variety of dietary intervention types for pain relief [35]. Given the high prevalence and increasing burden of OA [1,36], identifying the optimal treatment strategy for preventing sarcopenia is relevant for OA, especially for those who are overweight or obese.

Previous systemic review and meta-analysis studies have investigated the efficacy of diet alone or diet in combination with exercise for patients with OA and obesity [26,27,28,30,37], among which only two studies reported walk performance [26,27]; none of them focused on muscle mass and strength outcomes. In addition, few studies have compared the relative effects of combined treatments composed of different diet and exercise types. The purpose of this study was to (1) identify the relative effects of diet, exercise, and combined treatment on muscle mass, strength, and walking speed by using network meta-analysis (NMA) and (2) identify the optimal treatment by ranking the probability of each intervention type for people with overweight or obesity and knee or hip OA.

## 2. Materials and Methods

### 2.1. Design

The present NMA study was conducted in accordance with the guidelines of the Preferred Reporting Items for Systematic Reviews and Meta-Analyses [38] and the extension statement for reporting of systematic reviews incorporating NMA [39,40]. The study protocol was registered at PROSPERO (registration number: CRD42021198023). A comprehensive search of online sources was performed on 12 May 2021 to identify eligible randomized controlled trials (RCTs) examining the efficacy of diet, exercise, or combined treatment for patients with OA who had overweight or obesity. Articles were obtained from the following online databases: PubMed, EMBASE, the Physiotherapy Evidence Database (PEDro), the ClinicalKey database, the Cochrane Library Database, the China Knowledge Resource Integrated Database, and Google Scholar. Secondary sources included papers cited in articles retrieved from the aforementioned sources. To minimize publication and language biases, no limitation was imposed on the publication year or language. Two authors (CDL and HCC) independently searched for relevant articles, screened them, and extracted data. Any disagreement between the authors was resolved through consensus reached in collaboration with the other team members (SFC and THL).

### 2.2. Search Strategy

The following keywords were used for participants’ conditions: (“older adult” OR “elder individual”) AND (“overweight” OR “obesity”) AND “osteoarthritis”. The following keywords were used for the intervention: (“land exercise training” OR “water exercise training”) OR (“diet intervention” OR “weight loss”). The detailed search formulas for each database are presented in Appendix A.

### 2.3. Criteria for Selecting Studies

Trials were included if they met the following criteria: (1) the study design was an RCT or quasi-RCT; (2) the study enrolled participants who were aged ≥40 years, had body mass index (BMI) ≥25 kg/m^2^, and had a symptom or radiographic diagnosis of primary hip or knee OA. Participants were excluded if they had comorbidities such as rheumatic arthritis, neurological diseases (e.g., stroke, spinal stenosis), and substantial abnormalities in hematological, hepatic, or renal functions; (3) the treatment groups received diet therapy alone, exercise alone, or their combination; (4) the control group received a comparative intervention including placebo treatment alone or an intervention not related to diet therapy or exercise, which was considered usual care (UC) in the present study; (5) the diet therapy involved weight loss, dietary protein, or protein supplementation; (6) the exercise involved any mode of muscle strength therapy including resistance exercise training (RET), multicomponent exercise training (MET)—composed of two or more of RET, aerobic exercise (AET), balance training, and physical activity training—and an exercise intervention with physical therapies such as neuromuscular electric stimulation (NMES) and blood flow restriction (BFR); (7) the study conducted an acute intervention with a short period ranging from a few days to 12 weeks, a medium-term or a long-term intervention with a treatment period of ≥6 months; and (8) the study reported outcome measures including the primary and secondary outcomes defined below (in Section 2.4).

Studies were excluded if (1) the trial was conducted in vitro or in vivo in an animal model or (2) the trial had a non-RCT design such as case report, case series, or prospectively designed trial without a comparison group.

Study selection was initially performed by two authors (CDL and HCC) who independently screened and identified potentially relevant articles based on title and abstract. The full texts of all potentially eligible articles were examined to ensure they matched the inclusion criteria. In cases of inconclusiveness, the disagreements were resolved by discussions until consensus was obtained. A third author (THL) was consulted to discuss eligibility if the disagreement persisted.

### 2.4. Outcome Measures

The primary outcomes of interest included measures of muscle mass, strength, and walking speed, all of which are sarcopenia indicators recommended by the Asian Working Group for Sarcopenia [14] and the European Working Group on Sarcopenia in Older People [15]. The muscle mass measures included, but were not limited to, LBM, FFM, appendicular lean mass, muscle cross-sectional area, muscle volume, and muscle thickness. If LBM or FFM was not available, BF% and body weight was used to estimate the percentage FFM. For muscle strength measures, muscle quality (i.e., ratio of muscle strength to muscle mass) was prioritized [15]. Other strength measures were extracted on the basis of the following sequence of preference: concentric/eccentric power and peak torque, and maximum voluntary isometric contraction of knee extensors, knee flexors, and hip abductors; bench press and hand grip strength of upper extremity. Walking speed was assessed using gait and walking parameters (e.g., walking time).

### 2.5. Data Extraction

The following data were extracted from each included trial: (1) characteristics of the study design and sample, including study arm, age, BMI, and sex distribution; (2) characteristics of the exercise and diet interventions; (3) measurement time points; and (4) main outcomes. One author (CDL) extracted the relevant data from included trials, and the second author (HCC) checked the extracted data. Any disagreement between the two authors was resolved through the consensus procedure. A third author (THL) was consulted if the disagreement persisted.

Treatment effects of varying intensities or bilateral legs were combined to obtain a single treatment effect as recommended in the Cochrane Handbook [41]. The follow-up duration was assessed and was defined as short-term (≤3 months), medium-term (<3 months and ≤6 months), and long-term (>6 months) for subgroup analysis; when multiple time points were reported within the same time frame, the longest period was used for analysis (e.g., if the follow-up time points for walking speed measurement were 6 and 12 months, the data for the 12-month period were used as the long-term results). We also examined the compliance for interventions as well as adverse events reported by the included studies.

### 2.6. Assessment of Bias Risks and Methodological Quality of Included Studies

Quality was assessed using the PEDro quality score to determine the risk of bias. The methodological quality of all the included studies was independently assessed by two researchers in accordance with the PEDro classification scale, which is a valid measure of the methodological quality of clinical trials [42]. In the PEDro scale, the following 10 items are scored: random allocation, concealed allocation, similarity at baseline, subject blinding, therapist blinding, assessor blinding, >85% follow-up for at least one key outcome, intention-to-treat analysis, between-group statistical comparison of at least one key outcome, and point and variability measures for at least one key outcome. Each item is scored as either 1 for present or 0 for absent, and a total score ranging from 0 to 10 is obtained through summation of the scores of all the 10 items. An interrater reliability generalized kappa statistic value between 0.53 and 0.94 has been reported for the PEDro scale [43], and an intraclass correlation coefficient of 0.91 (95% confidence interval (CI): 0.84–0.95) associated with the cumulative PEDro score has been reported for nonpharmacological studies [44]. In this study, on the basis of the PEDro score, the methodological quality of the included RCTs was rated as high (≥7/10), medium (4–6/10), and low (≤3/10) [45].

### 2.7. Data Synthesis and Analysis

We separately computed effect sizes for each outcome measure in each intervention. The effect size was defined as a pooled estimate of the mean difference in change scores between any two study arms. Change scores (i.e., change from baseline) were analyzed to partially correct between-participant variability [46]. Change scores were extracted when the mean and standard deviation (SD) of the changes were available. If the exact variance of the paired difference was not derivable, conservative estimation was performed by assuming a within-participant correlation coefficient of 0.7, as recommended by Rosenthal (1993) [47], between the baseline and posttest measured data. If the SD was not reported, it was estimated using *p*-values or 95% CIs. If data were presented as the median and interquartile range, the mean value was estimated using the median, and the SD was calculated by dividing the interquartile range by 1.35 [46].

Because of the varied tools employed for measuring muscle mass, strength, and walking speed among the included studies, the standard mean difference (SMD) with 95% CI was calculated for all the extracted outcome data to ensure sufficient comparability of effect sizes. The odds ratio with 95% CI was estimated for dichotomous outcomes, such as occurrence rate of adverse events. We categorized the magnitude of the SMD using the following Cohen’s criteria [48]: trivial (*d* < 0.10), small (0.10 ≤ *d* < 0.25), medium (0.25 ≤ *d* < 0.40), and large (*d* ≥ 0.40).

We conducted random-effects NMA within a frequentist framework by using statistical software R (version 4.0.4, The R Foundation for Statistical Computing, Vienna, Austria) [49,50]. Direct and indirect comparisons of different diet and ET interventions were performed using the netmeta package of R [51]. Tests of heterogeneity (within designs) and inconsistency (between designs) were performed using the *I*^2^ statistic and Cochran Q test. The magnitude of τ^2^ was calculated to estimate the variance across all treatment comparisons. The design-by-treatment inconsistency model, loop-specific approach, and node-splitting method were used to assess the inconsistency between the direct and indirect comparisons [52,53]. A two-sided *p*-value of <0.05 was considered statistically significant.

For evaluation of relative efficacy, we calculated the ranking probabilities of each treatment on the basis of effect sizes by using the frequentist treatment ranking method [54]; and a P-score which has been shown to be equivalent to the surface under the cumulative ranking curve score within a Bayesian framework was expressed to represent the probabilities for each treatment [54]. Network forest plots for treatment effects in comparison with UC effects were produced to provide a visual representation of the uncertainty in NMA [55].

Network meta-regression analyses were performed to assess the moderating effects of age, BMI, sex distribution (i.e., proportion of female participants in the sample), treatment duration, sample size, and methodological level (i.e., PEDro score) on the relative efficacy of treatments.

Potential publication bias was investigated through visual inspection of a funnel plot to explore possible reporting bias [56] and was assessed using Egger’s regression asymmetry test [57].

## 3. Results

### 3.1. Trial Selection Flowchart

Figure 1 shows a flowchart of the trial selection process. Through an electronic and manual literature search, we identified 966 articles. After excluding duplicates, we reviewed the titles and abstracts of 248 studies to assess their eligibility; 91 were considered to be relevant for full-text assessment (Figure 1). The final sample consisted of 34 RCTs [58,59,60,61,62,63,64,65,66,67,68,69,70,71,72,73,74,75,76,77,78,79,80,81,82,83,84,85,86,87,88,89,90,91], which were derived from 28 trials and were published between 2000 and 2020. Two of the included RCTs [58,74] had a common study protocol [92], as did another two RCTs [59,62], two RCTs [82,83], and three RCTs [75,76,91], which respectively employed three trial protocols [76,93,94].

### 3.2. Study Characteristics

Table 1 summarizes the study characteristics of and patient demographic data from the included RCTs. A total of 3563 participants were recruited, with a mean age of 64.7 (range: 41.4–71.6) years; the mean BMI was 33.1 (range: 26.4–41.4) kg/m^2^, and the average proportion of female participants was 71.0% (range: 39.1–95.0%), which was estimated by excluding eight sex-specific (women or men alone) RCTs [68,69,71,77,80,81,88,89]. All included RCTs enrolled participants who had received a diagnosis of symptomatic or radiographic knee OA, whereas four RCTs recruited individuals with a diagnosis of hip OA [64,67,72,85].

In this study, 23 of the included RCTs had a two-arm design [61,64,65,66,67,68,69,73,75,76,77,78,80,81,82,83,84,85,86,87,89,90,91] and the remaining 11 RCTs were multiarm studies, with a total of 21 treatment arms. Among all participants, 711 (19.9%) received diet therapy alone, 1235 (34.7%) received exercise alone, 858 (24.1%) received combined treatment, and 759 (21.3%) received UC. Regarding the follow-up duration for measuring outcomes, 23 RCTs [61,63,64,65,66,67,68,69,70,71,72,77,79,80,81,82,84,85,86,87,88,89,90] had a short-term follow-up duration of ≤12 weeks, whereas 12 RCTs [58,64,73,74,75,76,78,82,85,86,87,91] had a medium-term follow-up duration ranging from 14 to 24 weeks, and six RCTs [58,69,70,74,82,83] had a long-term follow-up duration ranging from 8 to 36 months.

**Table 1 nutrients-13-01992-t001:** Summary of the study characteristics of included trials.

Study (Author, y, Ref)	Study Arm ^1^	Age (y) ^2^	BMI (kg/m^2^)	Sex (F/M)	N	Involved Joint	Exercise Intervention		Diet Intervention		Measured Time Point (weeks)	Main Outcome Measure
							Type (COM%)	Frequency × Duration	Type (COM%)	WL Target (% or kg) ^6^		
Diet intervention alone												
Christensen,	D	60.5 ± 11.6	36.3 ± 5.6	35/5	40	Knee	None	None	MR	None	Baseline	FFM
2005 [61]	UC	64.6 ± 10.4	35.5 ± 4.6	36/4	40				(6 meals/d),		Posttest: 8	
									DIA, CBT			
									−93.2			
Christensen,	D1, Regular MR	63.7 ± 6.5	32.6 ± 3.7	62/15	77	Knee	None	None	MR	None	Baseline	LBM
2017 [60]	D2, Intermittent	63.9 ± 6.3	34.0 ± 5.3	65/11	76				(1–3 meals/d),		Midtest: 52	
	MR								DIA		Posttest: 156	
									−71.2			
López-	D1, Uni-MR	60.9 ± 11.2	38.9 ± 5.1	35/17	52	Hip	None	None	MR (1 or	↓ 5 kg	Baseline	FFM
Gómez,	D2, Multi-MR	61.4 ± 11.0	40.2 ± 5.3	46/14	60	Knee			2 meals/d)		Posttest: 12	
2020 [67]									(NR)			
Exercise intervention alone												
Gill,	Water-based ET	69.2 ± 10.5	41.4 ± 3.9	28/14	42	Hip	AQET,	2 d/wk × 6 wk	None	None	Baseline	GS
2009 [64]	Land-based ET	71.6 ± 8.9	39.8 ± 13.1	23/17	40	Knee	RET	(12 sessions)			Posttest: 7	
							(81.7–87.5)				Follow-up: 15	
Kuptniratsaikul,	Water-based ET	62.1 ± 6.4	27.9 ± 1.5	38/2	40	Knee	AQET	3 d/wk × 4 wk	None	None	Baseline	Qd strength
2019 [65]	Land-based ET	61.7 ± 6.9	27.6 ± 1.7	37/3	40		IMET	(12 sessions)			Posttest: 4	
							(76–91.7)					
Lim,	Water-based ET	65.7 ± 8.9	27.9 ± 1.5	23/3	26	Knee	AQET;	3 d/wk × 8 wk	None	None	Baseline	LBM
2010 [66]	Land-based ET	67.7 ± 7.7	27.6 ± 1.7	21/4	25		RET;	(24 sessions)			Posttest: 8	FFM
	Home-based ET	63.3 ± 5.3	27.7 ± 2.0	21/3	24		(NR)					Qd strength
Mahmoud,	ET	54.6 ± 8.6	35.0 ± 4.1	0/32	32	Knee	IM-ET	3 d/wk × 12 wk	None	None	Baseline	MT
2017 [69]	UC ^3^	53.2 ± 9.6	34.8 ± 4.2	0/12	12		(NR)	(36 sessions)			Posttest: 12	Qd strength
Mangani,	AET	68.7 ± 5.6 ^5^	33.3 ± 5.0 ^5^	146/51 ^5^	57	Knee	AET	3 d/wk × 72 wk	None	None	Baseline	GS
2006 [70]	RET				64		RET	(216 sessions)			Mid-test: 12, 36	
	UC				76		(50.5–81.4)				Posttest: 72	
Matsuse,	NMES + ET	58.8 ± 11.8	37.5 ± 4.5	10/0	10	Knee	AET	2 d/wk × 12 wk	None	None	Baseline	GS
2020 [71]	ET	59.7 ± 6.1	36.1 ± 3.4	10/0	10		(NR)	(24 sessions)			Posttest: 12	Qd strength
Rabe,	NMES	67.3 ± 8.5	27.5 ± 6.1	17/0	17	Knee	RET	2 d/wk × 12 wk	None	None	Baseline	GS
2018 [77]	ET	65.9 ± 9.4	27.5 ± 4.1	18/0	18		(NR)	(24 sessions)			Posttest: 12	Qd strength
Exercise intervention alone												
Rosemffet,	NMES + ET	62.4 ± 8.7 ^4,5^	31.6 ± 5.3 ^4^	20/6 ^5^	8	Knee	MET	3 d/wk × 8 wk	None	None	Baseline	Qd strength
2004 [79]	NMES		34.2 ± 6.7 ^4^		8		−70.3	(24 sessions)			Posttest: 8	
	ET		29.1 ± 3.3 ^4^		10							
Segal,	BFR-ET	58.4 ± 8.7	31.3 ± 5.3	0/19	19	Knee	RET	3 d/wk × 4 wk	None	None	Baseline	Leg press 1-RM
2015a [80]	ET	56.1 ± 7.7	30.4 ± 4.2	0/22	22		−100	(12 sessions)			Posttest: 4	
Segal,	BFR-ET	56.1 ± 5.9	28.7 ± 4.4	19/0	19	Knee	RET	3 d/wk × 4 wk	None	None	Baseline	Qd volume
2015b [81]	ET	54.6 ± 6.9	32.5 ± 5.2	21/0	21		−97.2	(12 sessions)			Posttest: 4	Leg press 1-RM
Swank,	ET	63.1 ± 7.3	35.9 ± 8.5	24/12	36	Knee	RET	3 d/wk × 4–8 wk	None	None	Baseline	Qd strength
2011 [84]	UC ^3^	62.6 ± 7.6	32.9 ± 5.7	22/13	35		−90	(12–24 sessions)			Posttest: 8	
Tak,	ET	67.4 ± 7.6	26.4 ± 3.0	29/16	45	Hip	MET	7 d/wk × 8 wk	None	None	Baseline	GS
2005 [85]	UC ^3^	68.9 ± 7.6	26.6 ± 4.3	35/14	49		−77	(56 sessions)			Posttest: 8	
											Follow-up: 20	
Talbot,	ET	69.6 ± 6.7	31.0 ± 5.9	13/4	17	Knee	AET	3 d/wk × 12 wk	None	None	Baseline	GS
2003a [86]	UC ^3^	70.8 ± 4.7	32.6 ± 6.9	13/4	17		−76	(36 sessions)			Posttest: 12	Qd strength
											Follow-up: 24	
Talbot,	NMES	70.3 ± 5.6	29.5 ± 4.1	15/3	18	Knee	IMET	3 d/wk × 12 wk	None	None	Baseline	GS
2003b [87]	UC ^3^	70.8 ± 4.9	31.6 ± 5.9	4/12	16		−85	(36 sessions)			Posttest: 12	Qd strength
											Follow-up: 24	
Wallis,	ET	68.0 ± 8.0	34.0 ± 5.2	9/14	23	Knee	AET	7 d/wk × 12 wk	None	None	Baseline	GS
2017 [90]	UC ^3^	67.0 ± 7.0	34.0 ± 7.4	11/12	23		−70	(84 sessions)			Posttest: 13	
Combined treatments												
Beavers,	D + ET	65.5 ± 6.0	33.5 ± 3.7	108/43	151	Knee	MET	3 d/wk × 72 wk	MR	↓ ≥10%	Baseline	FFM ^4^
2015 [58]	ET	65.5 ± 6.4	33.5 ± 3.7	108/42	150		(58–70)	(216 sessions)	(2 meals/d),		Midtest: 24	
	D	65.8 ± 6.2	33.7 ± 3.8	105/44	149				DIA, CBT		Posttest: 72	
									−63			
Christensen,	D + ET	62.9 ± 5.8	36.5 ± 4.4	52/12	64		MET	3 d/wk × 52 wk	MR	↓ ≥10%	Baseline	LBM;
2013 [59];	D	63.0 ± 6.5	37.9 ± 5.3	52/12	64		−59.1	(156 sessions)	(1 meal/d),		Posttest: 52	
2015 [62]	UC	61.7 ± 6.8	37.6 ± 4.5	51/13	64				DIA, CBT			
									−61.5			
Combined treatments												
Ghroubi,	D + ET	41.4 ± 3.9	37.5 ± 3.7	NR	12	Knee	MET	3 d/wk × 8 wk	MR	NR	Baseline	LBM
2008 [63]	ET	43.8 ± 13.1	37.1 ± 5.7		10		(NR)	(24 sessions)	(3 meals/d),		Posttest: 8	
	D	41.5 ± 11.7	38.7 ± 6.1		12				(NR)			
	UC ^3^	42.4 ± 9.8	39.2 ± 3.7		11							
Magrans-	D + ET	54.0 ± 9.0 ^5^	33.3 ± 5.0 ^5^	14/0	14	Knee	RET	3 d/wk × 14 wk	DIA	↓ 3–5 kg	Baseline	FFM
Courtney,	ET			16/0	16		(NR)	(42 sessions)	(NR)		Midtest: 10	Bench-
2011 [68]											Posttest: 14	press 1-RM
McLeod,	D + ET	66.5 ± 4.8	33.7 ± 7.6	64/8	72	Hip	MET	3 d/wk × 8 wk	DIA, CBT	↓ 5%	Baseline	LBM
2020 [72]	ET	67.2 ± 5.7	33.9 ± 7.3	73/10	83	Knee	(NR)	(24 sessions)	(NR)		Posttest: 8	FFM ^4^
Messier,	D + ET	67.0 ± 4.0	35.0 ± 5.0	10/3	13	Knee	MET	3 d/wk × 24 wk	DIA, CBT	↓ ≥6.8 kg	Baseline	GS
2000 [73]	ET	69.0 ± 5.0	38.0 ± 6.0	7/4	11		−94.7	(72 sessions)	−82.6		Midtest: 12	Qd strength
											Posttest: 24	
Messier,	D + ET	65.0 ± 6.0	33.6 ± 3.7	109/43	152	Knee	MET	3 d/wk × 72 wk	MR	↓ 10–15%	Baseline	LBM
2013 [74]	ET	66.0 ± 6.0	33.5 ± 3.7	108/42	150		(54–70)	(216 sessions)	(2 meals/d),		Midtest: 24	
	D	66.0 ± 6.0	33.7 ± 3.8	108/44	152				DIA, CBT		Posttest: 72	
									(61.0–63.0)			
Miller,	D + ET	69.8 ± 8.2	34.9 ± 6.5	43/24	67	Knee	MET	3 d/wk × 24 wk	MR,	↓ 10%	Baseline	FFM
2006 [76]	UC ^3^	69.5 ± 8.2	34.4 ± 5.7	38/29	67		(72–83)	(72 sessions)	(2 meals/d)		Posttest: 24	
									DIA, CBT			
									−75			
Miller,	D + ET	69.3 ± 6.6	35.7 ± 6.6	14/12	26	Knee	MET	3 d/wk × 24 wk	MR	↓ 10%	Baseline	FFM
2012 [75]	UC ^3^	69.3 ± 6.5	34.9 ± 4.0	13/12	25		−76.3	(72 sessions)	(2 meals/d),		Posttest: 24	
									DIA, CBT			
									−74			
Robbins,	D + ET	62.5 ± 7.4	34.6 ± 6.9 ^4^	57/30	87	Knee	MET	3 d/wk × 18 wk	MR	NR	Baseline	GS
2020 [78]	UC ^3^	63.8 ± 7.3	36.3 ± 7.5 ^4^	52/32	84		(89/73)	(54 sessions)	(2 meals/d),		Midtest: 20	Qd strength
									−94		Posttest: 32	
Skou,	D + ET	64.8 ± 8.7	30.6 ± 5.6	26/24	50	Knee	MET	2 d/wk × 12 wk	DIA	↓ ≥5%	Baseline	GS
2015 [82]	UC ^3^	67.1 ± 9.1	29.4 ± 5.2	25/25	50		−65.8	(24 sessions)	−67.5		Posttest: 12	
											Follow-up: 24, 52	
Combined treatments												
Skou,	D + ET	65.9 ± 8.7	31.3 ± 5.7	56/44	100	Knee	MET	2 d/wk × 12 wk	DIA	↓ ≥5%	Baseline	GS
2018 [83]	UC ^3^	67.1 ± 9.1	29.4 ± 5.2	25/25	50		(NR)	(24 sessions)	(NR)		Posttest: 12	
											Follow-up: 24, 104	
Toda,	D + ET	56.5 ± 11.1	28.1 ± 1.1	11/0	11	Knee	AET	7 d/wk × 8 wk	MR	NR	Baseline	LBM
2000 [89]	UC ^3^	61.9 ± 5.5	28.8 ± 3.3	31/0	31		RET	(56 sessions)	(2 meals/d),		Posttest: 8	
							(NR)		(NR)			
Toda,	D + ET	63.2 ± 7.9	27.4 ± 3.1 ^4^	63/0	63	Knee	AET	7 d/wk × 8 wk	MR	NR	Baseline	LLM
2001 [88]	ET	61.0 ± 11.7	26.6 ± 4.0 ^4^	84/0	84		RET	(56 sessions)	(2 meals/d),		Posttest: 8	
	D	60.1 ± 13.5	27.9 ± 4.7 ^4^	29/0	29		(NR)		(NR)			
	UC ^3^	63.1 ± 9.3	26.4 ± 4.1 ^4^	52/0	52							
Wang,	D + ET	69.9 ± 5.7	35.0 ± 5.8	25/15	40	Knee	MET	3 d/wk × 24 wk	MR	↓ 10%	Baseline	LBM
2007 [91]	UC ^3^	68.8 ± 5.7	34.7 ± 4.3	21/12	33		−77.5	(72 sessions)	(2 meals/d),		Posttest: 24	Qd strength
									DIA, CBT			
									−75			

^1^ All study arms are presented for each trial. ^2^ Values are presented as mean and SD (or range). ^3^ No diet intervention, and no muscle strength exercise training. ^4^ Data were estimated. ^5^ Values of all samples. ^6^ Values denote the weight loss target in terms of body weight reduction in % or kg. AET, aerobic exercise training; AQET, aquatic exercise training; BFR-ET, exercise training with blood flow restriction; BMI, body mass index; CBT, cognitive behavioral therapy; COM, compliance; D, diet; DIA, diet instruction and advisement; ET, exercise training; FFM, fat-free mass; GS, gait speed; IMET, isometric exercise training; LBM, lean body mass; LLM, leg lean mass; MET, multicomponent exercise training; MR, meal replacement; MT, muscle thickness; NC, nutrition class; NMES, neuromuscular electric stimulation; NR, not reported; PLA, placebo; Qd; quadriceps muscle; Ref, reference number; RET, resistance exercise training; UC, usual care; WL, weight loss.

### 3.3. Dietary Intervention Characteristics

The protocols for diet therapy are summarized in Table 1. Dietary interventions employed intermittent meal replacement (MR) in one RCT [60] and regular MR in 14 RCTs [58,59,60,61,62,63,67,74,75,76,78,88,89,91]. The MR prescribed was one meal daily (uni-MR) in four RCTs [59,60,62,67] and two or more meals daily (multi-MR) in 12 RCTs [58,60,61,63,67,74,75,76,78,88,89,91]. In addition, diet instruction and advisement (DIA), which had been conducted through nutrition classes and cognitive behavior therapy, was employed in five RCTs [68,72,73,82,83] or applied in combination with MR in nine RCTs [58,59,60,61,62,74,75,76,91]. In summary, a total of five types of diet therapy for weight management were included in the NMA: intermittent multi-MR with DIA (IMMR-DIA), regular uni-MR (RUMR), RUMR combined with DIA (RUMR-DIA), regular multi-MR (RMMR), and RMMR combined with DIA (RMMR-DIA).

### 3.4. Exercise Training Protocol

A summary of protocols for exercise is presented in Table 1. Regarding the training mode of exercise, seven types of exercise were identified; AET was used in six RCTs [70,71,86,88,89,90] and RET in 10 RCTs [64,66,68,70,77,80,81,84,88,89], aquatic exercise (AQET) in three RCTs [64,65,66], isometric exercise in three RCTs [65,66,69], MET in 13 RCTs [58,59,62,63,72,73,74,75,76,78,79,85,91], NMES in four RCTs [71,77,79,87], and RET with BFR in two RCTs [80,81]. Moreover, 23 RCTs [58,59,62,63,64,65,66,68,69,70,71,73,74,77,79,80,81,84,85,86,87,88,90] employed exercise alone, whereas 14 RCTs incorporated exercise with dietary interventions [58,63,68,72,73,74,75,76,78,82,83,88,89,91]. Regarding the treatment duration, 19 RCTs [63,64,65,66,69,71,72,77,79,80,81,82,83,84,85,86,87,88,89] conducted a short intervention lasting 4–12 weeks (12–56 sessions), whereas eight RCTs [68,70,73,75,76,78,90,91] had a medium-term exercise duration of 14–24 weeks (42–72 sessions); in addition, five RCTs [58,59,62,70,74] had a long-term training period of 52–72 weeks (156–216 sessions).

### 3.5. Risk of Bias in Included Studies

The individual PEDro scores are listed in Appendix A. Overall, methodological quality assessment revealed that half of the included RCTs [59,60,62,64,65,66,68,69,74,78,80,81,82,83,85,88,90] had high methodological quality and the other 17 RCTs [58,61,63,67,70,71,72,73,75,76,77,79,84,86,87,89,91] were ranked as medium, with a median PEDro score of 6.5/10 (range: 4/10 to 8/10). The interrater reliability of the cumulative PEDro scores was acceptable, with an intraclass correlation coefficient of 0.97 (95% CI: 0.93–0.98). All the 34 included RCTs employed random allocation, similarity at baseline, between-group comparisons, and point estimates and variability. Moreover, 11 of the 17 (64.7%) high-quality RCTs [59,60,62,64,65,78,80,81,82,83,90] performed allocation concealment, whereas only 1 medium-quality RCT did [71]. Owing to the intervention nature, it was difficult to blind the participants and therapists in all the included RCTs. However, assessor blinding was performed by all the high-quality RCTs, as well as five medium-quality RCTs [58,70,71,73,77].

### 3.6. Effectiveness of Treatment for Muscle Mass Assessed in NMA

Figure 2 presents the network of eligible comparisons for the treatment options for patients with overweight or obesity and OA. The effects of each treatment relative to UC on muscle mass, muscle strength, and walking speed at each follow-up time point are shown in Figure 3, Figure 4 and Figure 5, respectively, and the details of each comparison are presented in Appendix A. The supplementary league tables, Appendix A, present direct comparisons from the pairwise meta-analysis and NMA results and the relative efficacy of different treatments in comparison with UC.

#### 3.6.1. Pairwise Meta-Analysis

Direct comparisons of pairwise meta-analyses (Appendix A) indicated that IMET (SMD = 1.13, 95% CI: 0.27–1.98) and regular IMMR-DIA (SMD = 1.02, 95% CI: 0.35–1.69) were more efficacious than UC for increasing muscle mass, as were the combined treatments regular IMMR-DIA plus MET (SMD = 1.16, 95% CI: 0.75–1.58) and regular multi-MR plus either RET (SMD = 1.26, 95% CI: 0.59–1.92) or AET (SMD = 1.17, 95% CI: 0.47–1.86). In addition, the combined treatment of regular multi-MR plus RET produced greater changes in muscle mass than regular multi-MR alone (SMD = 0.76, 95% CI: 0.19–1.33) or RET alone (SMD = 0.64, 95% CI: 0.01–1.29).

#### 3.6.2. Global Effects in NMA

The NMA results showed that in comparison with UC, the diet interventions RMMR-DIA (SMD = 0.79) and RMMR (SMD = 0.51) produced greater changes in muscle mass—as did the exercise interventions AQET (SMD = 1.04), RET with BFR (SMD = 1.00), IMET (SMD = 0.79), RET (SMD = 0.63), and MET (SMD = 0.43)—during the overall follow-up duration (Figure 3 and Appendix A). In addition, the combined effects of RMMR plus RET (SMD = 1.40), RMMR-DIA plus MET (SMD = 0.99), RMMR plus AET (SMD = 0.93), and DIA plus MET (SMD = 0.69) on muscle mass gain relative to UC appeared to be stronger than the effect of diet intervention alone or exercise alone. Pooling all treatment effects in NMA, RMMR plus RET was ranked as the most effective (P score = 0.95) among all treatment arms for muscle mass gain—followed by regular IMMR-DIA plus MET (P score = 0.78), AQET (P score = 0.77), and RET with BFR (P score = 0.73)—during the overall follow-up duration (Figure 3 and Appendix A). The global heterogeneity of the NMA model for muscle mass was significant (*τ*^2^ = 0.06, *I*^2^ = 56.7%, *p* = 0.005). The node splitting results for inconsistency of NMA revealed no inconsistencies between direct and indirect evidence; the same results were detected through visual inspection of the forest plot (Appendix A).

#### 3.6.3. Subgroup Analysis of Follow-Up Duration

The combined treatment RMMR plus RET was ranked as the optimal treatment for muscle mass (SMD = 1.44, 95% CI: 0.57–2.31, P score = 0.88) over the short-term follow-up duration, whereas RMMR-DIA plus MET (SMD = 0.50, 95% CI: 0.11–0.90, P score = 0.97) was ranked highest over the long-term follow-up (Figure 3 and Appendix A); only a single comparison was conducted in NMA over medium-term follow-up, and the results showed that RMMR-DIA plus MET exerted a significant effect on muscle mass (SMD = 1.16, 95% CI: 0.86–1.47).

### 3.7. Effectiveness of Treatment for Muscle Strength

#### 3.7.1. Pairwise Meta-Analysis

Direct comparisons of pairwise meta-analyses (Appendix A) indicated that AET (SMD = 0.51, 95% CI: 0.10–0.91) and MET (SMD = 0.42, 95% CI: 0.20–0.65) were more efficacious than UC for muscle strength, as was DIA plus MET (SMD = 0.52, 95% CI: 0.26–0.77). In addition, the combined treatment RMMR-DIA plus MET yielded greater changes in walking speed than RMMR-DIA alone (SMD = 1.64, 95% CI: 1.29–1.99) or MET alone (SMD = 0.55, 95% CI: 0.22–0.87); similar results were observed for the comparison between DIA plus MET and DIA alone (SMD = 0.68, 95% CI: 0.26–1.11).

#### 3.7.2. Global Effects of NMA

The NMA results showed that during the overall follow-up duration, the exercise interventions RET (SMD = 1.93), RET with BFR (SMD = 2.03), and NMES (SMD = 1.18) exerted significant effects on strength gain relative to UC, whereas the diet intervention and combined treatment did not (Figure 4 and Appendix A). RET was ranked as the most effective (P score = 0.82) among all treatment arms for muscle strength—followed by RET with BFR (P score = 0.81), NMES plus MET (SMD = 1.27, P score = 0.61), and AET (SMD = 1.23, P score = 0.60)—during the overall follow-up duration (Figure 4 and Appendix A). The global heterogeneity of the NMA model for muscle strength was significant (*τ*^2^ = 0.16, *I*^2^ = 74.2%, *p* < 0.0001). The node splitting results for inconsistency of NMA showed no inconsistencies between direct and indirect evidence; the same results were detected through visual inspection of the forest plot (Appendix A).

#### 3.7.3. Subgroup Analysis of Follow-Up Duration

RET plus BFR and NMES ranked the highest in terms of short-term (SMD = 2.06, 95% CI: 0.28–3.84, P score = 0.79) and medium-term (SMD = 0.71, 95% CI: 0.01–1.41, P score = 0.79) treatment efficacy, respectively, for muscle strength (Figure 4 and Appendix A); only one RCT [78] reported the long-term treatment effects of RMMR plus MET (SMD = 0.15, 95% CI: −0.17 to 0.46).

### 3.8. Effectiveness of Treatment for Walking Speed

#### 3.8.1. Pairwise Meta-Analysis

Direct comparisons of pairwise meta-analyses (Appendix A) indicated that AET (SMD = 0.47, 95% CI: 0.15–0.80) was more efficacious than UC for the walking speed, as well as the combined treatment DIA plus MET (SMD = 0.36, 95% CI: 0.02–0.69) did. In addition, the combined treatment DIA plus MET yielded greater changes in walking speed than MET alone (SMD = 1.12, 95% CI: 0.21–2.03).

#### 3.8.2. Global Effects of NMA

The NMA results revealed that during the overall follow-up duration, AET alone (SMD = 0.46) and DIA plus MET (SMD = 0.45) achieved greater changes in walking speed in comparison with UC (Figure 5 and Appendix A). In addition, during the overall follow-up duration, AET was ranked as the most effective (P score = 0.77) among all treatment arms regarding the effect on walking speed, followed by DIA plus MET (P score = 0.75), NMES plus AET (SMD = 0.46, P score = 0.66), and AQET (SMD = 0.28, P score = 0.55; Figure 5 and Appendix A). The global heterogeneity of the NMA model for muscle strength was insignificant (*τ*^2^ = 0.02, *I*^2^ = 20.1%, *p* = 0.28). The node splitting results for inconsistency of NMA showed no inconsistencies between direct and indirect evidence; the same result was detected through visual inspection of the forest plot (Appendix A).

#### 3.8.3. Subgroup Analysis of Follow-Up Duration

For walking speed, AET alone was discovered to be the optimal treatment in the short-term (SMD = 0.52, 95% CI: 0.25–0.80, P score = 0.87), medium-term (SMD = 0.55, 95% CI: 0.05–1.05, P score = 0.82), and long-term (SMD = 0.51, 95% CI: 0.16–0.86, P score = 0.92) follow-up durations, respectively (Figure 5 and Appendix A).

### 3.9. Network Meta-Regression Results for Moderators of Treatment Efficacy

The results of network meta-regression analyses are shown in Appendix A. No moderator was identified to have an influence on the treatment efficacy regarding muscle mass, strength gains, and walking speed.

### 3.10. Side Effects and Compliance

No serious adverse events, side effects, or severe complications were reported after diet therapy, exercise intervention, or combined treatment in all of the included RCTs. Nonserious adverse events related to exercise intervention were observed by nine RCTs [59,62,65,66,71,74,78,82,90], among which most commonly conditions were training-induced knee pain or muscle soreness of short duration (Appendix A). Three RCTs [60,61,78] reported adverse events related to the diet therapy which included food intolerances and mild gastrointestinal reactions (Appendix A). In addition, among the 16 RCTs [59,60,61,62,64,65,66,71,74,75,76,77,78,82,90,91] which had information about adverse events, seven [59,62,74,75,76,82,91] and five [64,75,76,77,91] reported no any adverse event occurred during (or after) diet therapy and exercise intervention, respectively.

The rate of compliance with the exercise interventions was 50.5–100% among the included RCTs that reported the adherence to exercise protocols or attendance rate of exercise sessions (Table 1) [58,59,62,65,70,73,74,75,76,78,80,81,82,84,86,87,91,95,96]. The rate of compliance with diet therapy was 61.0–94.7% in 13 RCTs [58,59,60,61,62,73,74,75,76,78,82,91,95].

### 3.11. Publication Bias

Visual inspection of the funnel plot of publication bias across the included RCTs for each primary outcome revealed no substantial asymmetry (Appendix A). Egger’s test results for muscle mass also did not indicate any obvious reporting bias among the RCTs included in the NMA (*p* = 0.24; Appendix A), nor did those for muscle strength (*p* = 0.42; Appendix A) or walking speed (*p* = 0.39; Appendix A).

## 4. Discussion

The primary goal of this study was to identify the relative efficacy of different diet therapies, exercise interventions, and combined treatments for muscle mass, strength, and functional outcomes in individuals with obesity and OA. The NMA results in the present study showed that (1) diet therapy alone (particularly RMMR) and exercise therapy alone had significant effects on muscle mass relative to UC, whereas combined treatment had additional treatment efficacy, irrespective of the specific intervention and follow-up duration; (2) in comparison with UC, exercise therapy alone and combined treatment exerted favorable effects on muscle strength and walking speed, respectively; (3) based on the cumulative ranking results, RMMR plus RET, RET alone, and AET alone were the optimal treatment strategy for muscle mass gain, strength gain, and walking speed recovery, respectively.

The present NMA can be considered clinically useful specifically because of the vast number of available treatment strategies and compositions for overweight or obese people who have lower-extremity OA. In the conservative pairwise meta-analysis coupling with multiple independent head-to-head trials, it is difficult to determine which treatment is the most efficacious [97]. By contrast, NMA provides consistent estimates of the relative treatment effects compared with each other using both direct (i.e., conservative pairwise meta-analysis) and indirect evidence without double counting the participants [98]. In addition, with a specific treatment such as RMMR plus MET, the inclusion of active comparators (i.e., RMMR alone or MET alone) is important as it encourages optimal clinical decision among different treatment options by clinicians for overweight or obese patients with OA. Even in the absence of head-to-head RCTs, this NMA provides clinicians with the bottom-line knowledge regarding the best available evidence on the comparative efficacy among different diet, exercise, and combined treatments for overweight or obese people with OA, especially those who are undergoing weight management.

In the present study, combined treatment incorporating RMMR and RET was overall ranked as the most effective treatment for muscle mass gain for a number of possible reasons. First, weight management may synergically cause decreases in absolute lean mass as well as total body weight and fat mass in older individuals with obesity and OA [31]. However, the negative effects of diet-induced weight loss on muscle mass may be overestimated when using changes in absolute LBM or FFM. In this NMA, the ratio of muscle mass to total body weight in terms of percentage FFM or LBM was used to estimate the effect size of each treatment, and the NMA results showed that diet intervention alone, particularly RMMR and RMMR-DIA, exerted positive effects on muscle mass gain relative to UC. Our findings indicate that the efficacy of diet therapy for muscle mass can be objectively estimated using percentage LBM or FFM rather than absolute values. Second, previous systemic review studies have indicated that muscle strength training, especially RET, augments muscle mass gain in older people with sarcopenia [99,100], individuals with obesity and OA [101], and older adults with OA [24]. The muscle mass gain caused by exercise may be beneficial for older adults with obesity who are undergoing weight management [31,102]. The previous results support our findings that RET plus RMMR for dietary weight management exerts superior effects on muscle mass relative to RMMR or RET alone. Our findings also indicate that muscle mass can be effectively maintained or muscle mass loss can be prevented by the adjunct therapy of RET in individuals with obesity and OA who are undergoing a dietary weight loss intervention. Finally, a previous NMA indicated that RET and MET are equally ranked as optimal exercise therapies for muscle mass gain in adults with obesity [101], which supports the present study finding that RET and MET combined with diet therapy were ranked as the first two optimal treatments for individuals with obesity and knee or hip OA. We further identified that PMMR plus RET had the highest efficacy for muscle mass gain over a short-term follow-up, and PMMR-DIA plus MET had the strongest long-term effect on muscle mass and walking speed outcomes.

The present NMA demonstrated that RET alone and RET with BFR were overall the two optimal treatments for muscle strength gain in people with obesity and OA; in addition, over the medium-term follow-up, NMES and AET had higher effects on muscle strength gain than combined treatment did. In OA, muscle weakness has been associated with low muscle mass [103]. Based on the facts that reduction in fat mass is accompanied by loss of absolute FFM caused by weight loss [29,30,31] and that decreased muscle mass is more sensitive for detecting muscle strength loss than fat mass reduction in individuals with OA [9], an exercise intervention employed as monotherapy may achieve more positive effects on muscle strength gain than dietary therapy alone, supporting the results in the current NMA.

In this NMA, a series of meta-regression analyses were performed to identify the factors affecting treatment efficacy, and we found no significant moderation effects of age, BMI, gender, sample size, and methodological level (i.e., PEDro score). However, substantial moderating effects of some factors were noted. First, substantially but insignificantly minor treatment effects were exerted on muscle strength and walking speed in participants of an older age. In addition, a similar association was observed between age and muscle mass gain, although the finding was nonsignificant (Appendix A). Previous studies have indicated that age-related changes in muscle mass are associated with a decline in strength and walking speed [104,105]. Conversely, intervention-induced increases in muscle mass may improve the strength and walking ability of older people who have sarcopenia and frailty risks [106]. The previous results may explain the parallel associations between age and muscle mass change and between age and strength and walking speed that were discovered in this NMA. Second, compared with male sex, female sex may be associated with poorer walking speed outcomes after interventions in such patients with obesity and OA. Results in this NMA were in line with a previous systemic review which found such a sex-specific difference in physical mobility after a diet plus exercise intervention in older individuals with OA [107]. Our findings further indicate that sex mediates the relative treatment efficacy among different diet therapies, exercise interventions, and combined treatments for physical mobility, which may be explained by the sex-specific muscle adaptations in response to diet and exercise interventions for the older population [108,109,110] and those with OA [24,111].

The findings of the present NMA should be interpreted on the basis of the following limitations. First, given the variation in dietary prescriptions and instructions (e.g., contents of MR and prescribed amounts, nutritional classes, and cognitive behavior therapy) and exercise protocols (training mode, duration, and volume), making a definite conclusion regarding the effect of specific types of treatment on muscle mass or strength gains was difficult. Second, some of our included trials had small group samples of fewer than 20 participants [63,68,69,71,73,77,79,86,87,88,96]; these studies found nonsignificant treatment effects on primary outcomes, which may have contributed negatively to the overall effect size. Third, because of the small number of RCTs with multiple exercise intensities (e.g., low and high intensities of RET), we combined the treatment effects of different exercise intensities within an exercise class, and the results should therefore be interpreted with caution. Fourth, the estimates for treatment arms—including IMMR, RUMR, AQET, NMES, RET with BFR, DIA plus RET, and DIA plus MET—were subject to considerable uncertainty with wide credibility intervals because the number of relevant RCTs was small. Finally, inadequate statistical power to detect inconsistency was noted owing to the small number of study arms relative to the number of treatment comparisons, although inconsistency was not detected in the current NMA.

## 5. Conclusions

This NMA determined the relative efficacy of different diet therapies, exercise interventions, and combined treatments for sarcopenia indices in individuals with obesity and hip or knee OA; in addition, the combined treatment RMMR plus RET was determined to be the optimal treatment strategy for muscle mass gain or preservation, whereas RET and AET alone were the optimal treatment options for strength and walking speed, regardless of the intervention and follow-up duration. According to the results of this study, we conclude that exercise alone increases muscle strength and walking speed, whereas an intervention incorporating diet therapy and exercise, especially RMMR plus RET, has superior effects on muscle mass and in individuals with obesity and lower-limb OA. The results of this study contribute to the knowledge on optimal diet and exercise intervention strategies, and an interdisciplinary and practical approach is required to counteract muscle loss and functional decline in the older population with obesity and OA. The findings of this review may guide the prescription of diet and exercise type for ensuring optimal treatment outcomes. Given the limitations of the current study, additional studies with large samples should be conducted for the identification of specific supplementation protocols.

## Figures and Tables

**Figure 1 nutrients-13-01992-f001:**
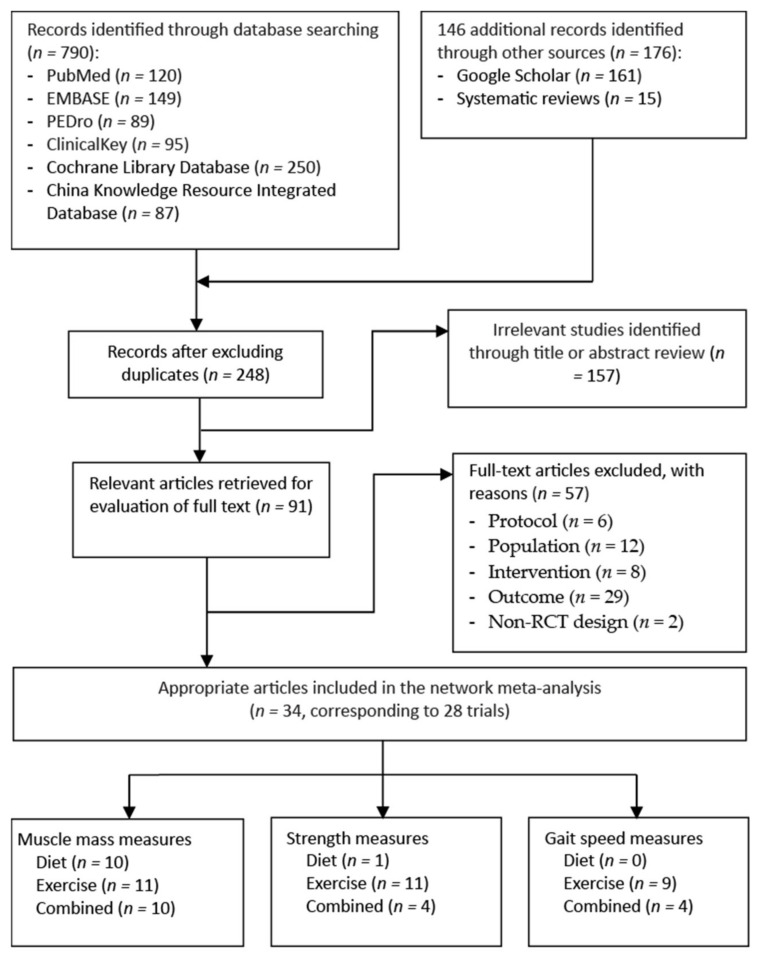
Flowchart of study selection.

**Figure 2 nutrients-13-01992-f002:**
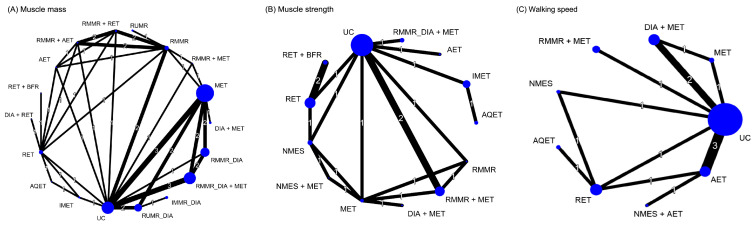
Network plot of direct comparisons for different treatments for (**A**) muscle mass, (**B**) muscle strength, and (**C**) walking speed. The lines between nodes represent direct comparisons in various studies; the thickness of each line is proportional to the number of studies denoted on the line and indicates the connection between studies in terms of comparisons. The size of each node is proportional to the sample size of the participants involved in each specific treatment. AET, aerobic exercise training; AQET, aquatic exercise training; BFR, blood flow restriction; DIA, diet instruction and advisement; IMET, isometric exercise training; IMMR, intermittent multimeal replacement; MET, multicomponent exercise training; NMES, neuromuscular electric stimulation; RET, resistance exercise training; RMMR, regular multimeal replacement; RUMR, regular unimeal replacement; UC, usual care.

**Figure 3 nutrients-13-01992-f003:**
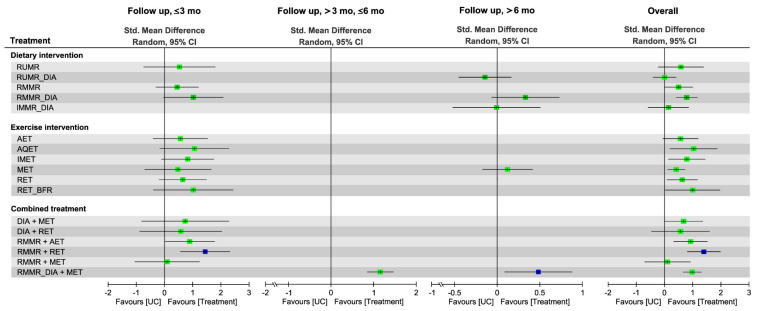
Forest plot summarizing the effects of diet, exercise intervention, and combined treatment on changes in muscle mass in at each follow-up time point. Each point estimate (square) in each time frame and during the overall duration presents the network combined effect (SMD) on the muscle mass relative to UC, with the 95% CI (horizontal line). The results plotted on the right-hand side indicate effects favoring the treatment approach. The blue−colored point denotes the highest rank of probability, reflecting that the treatment approach is the optimal intervention among all treatments in each time frame. 95% CI, 95% confidence interval; AET, aerobic exercise training; AQET, aquatic exercise training; BFR, blood flow restriction; DIA, diet instruction and advisement; IMET, isometric exercise training; IMMR, intermittent multimeal replacement; MET, multicomponent exercise training; RET, resistance exercise training; RMMR, regular multimeal replacement; RUMR, regular unimeal replacement; Std, standard; UC, usual care.

**Figure 4 nutrients-13-01992-f004:**
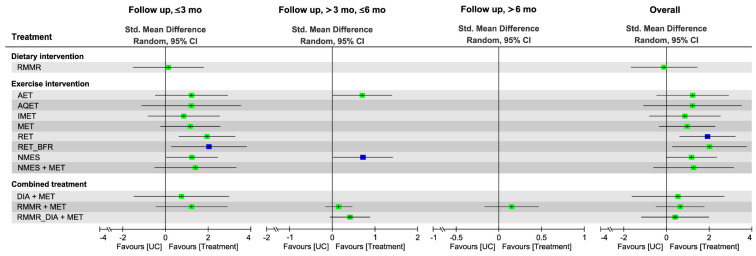
Forest plot summarizing the effects of diet, exercise intervention, and combined treatment on changes in muscle strength in each follow-up duration. Each point estimate (square) in each time frame and during the overall duration presents the network combined effect (SMD) on the muscle strength relative to UC, with the 95% CI (horizontal line). The results plotted on the right-hand side indicate effects favoring the treatment approach. The blue–colored point denotes the highest rank of probability, reflecting that the treatment approach is the optimal intervention among all treatments in each time frame. 95% CI, 95% confidence interval; AET, aerobic exercise training; DIA, diet instruction and advisement; MET, multicomponent exercise training; RET, resistance exercise training; RMMR, regular multimeal replacement; RUMR, regular unimeal replacement; Std, standard; UC, usual care.

**Figure 5 nutrients-13-01992-f005:**
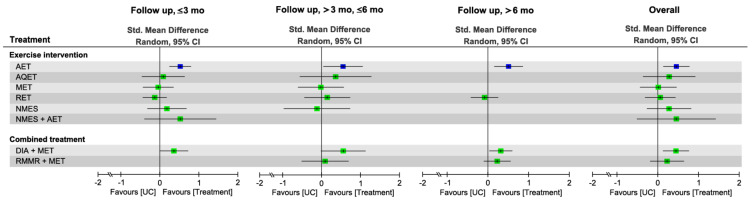
Forest plot summarizing the effects of diet, exercise intervention, and combined treatment on changes in walking speed in each follow-up duration. Each point estimate (square) in each time frame and during the overall duration presents the network combined effect (SMD) on the walking speed relative to UC, with the 95% CI (horizontal line). The results plotted on the right-hand side indicate effects favoring the treatment approach. The blue-colored point denotes the highest rank of probability, reflecting that the treatment approach is the optimal intervention among all treatments in each time frame. 95% CI, 95% confidence interval; AET, aerobic exercise training; AQET, aquatic exercise training; DIA, diet instruction and advisement; IMET, isometric exercise training; MET, multicomponent exercise training; NMES, neuromuscular electric stimulation; RET, resistance exercise training; RMMR, regular multimeal replacement; RUMR, regular unimeal replacement; Std, standard; UC, usual care.

## Data Availability

Refer to Appendix A. Raw data available on request.

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
