# Peer review of "Relative Efficacy of Weight Management, Exercise, and Combined Treatment for Muscle Mass and Physical Sarcopenia Indices in Adults with Overweight or Obesity and Osteoarthritis: A Network Meta-Analysis of Randomized Controlled Trials"

_nutrients, 2021, doi:10.3390/nu13061992_

Round 1

Reviewer 1 Report

In the article “Relative efficacy of diet, exercise, and combined treatment for muscle mass and physical sarcopenia indices in adults with overweight or obesity and osteoarthritis: A network meta-analysis of randomized controlled trials” the authors aimed to (1) identify the relative effects of diet, exercise, and combined treatment on muscle mass, strength, and walking speed by using network meta-analysis (NMA) and (2) identify the optimal treatment by ranking the probability of each intervention type for people with overweight or obesity and knee or hip OA.

While the NMA might be of interest to a broader audience, I have severe concerns about the transparency of the publication strategies of the authors.

Major concerns:

The authors have published a previous study https://www.mdpi.com/2072-6643/12/8/2422/htm under an identical protocol number at Prospero and a similar study aim: “Thus, this study examined the effects of PS + ET on muscle mass and functional outcomes in older adults with OA in the lower extremity”.

In my opinion, this is of ethical relevance as the current study has not been registered at PROSPERO as stated in the article and the authors did not openly report differences/similarities between the studies. Therefore, this issue needs to be clarified before judging about its suitability for publication.  

Please find below my suggestions for the manuscript, once the above issue has been adequately addressed.

Title:

I am not sure whether diet intervention is best suited in the context of your work. Did you address weight loss strategies, weight loss interventions, weight management …? This could be a bit more suitable (also to discriminate from protein interventions) and would fit at least to the information in your main table 1.

I do not have any suggestions for the introduction, which is easy to read and follows a logical argumentation.

Methods – there are some linguistic inaccuracies, which have to be addressed in more detail:

-) Please provide the data bases also in the main text, i.e. Google scholar is mentioned in Figure 1, but not in the supplementary table S1, where various data bases are described.

-) You mentioned the differences between short/medium/ long-term therapies. Did you exclude acute interventions? Were there any restrictions according to age of participants? What’s about comorbidities? Please clarify in the inclusion/exclusion criteria.

-) Who decided about inclusion/exclusion of a study? Was it similar to the data extraction process?

-) If LBM or FFM was not available, BF% was used to estimate the percentage FFM? You would need body mass/weight as well.

-) strength measures: knee extensors, knee flexors, and hip abductors are muscle groups not strength measures. What’s about power, peak torque, isometric/concentric/eccentric? What’s about upper extremities (ie. Bench press, which was the main outcome in Magrans-Courtney et al. (2011)). One point is maybe the most critical issue here: 6MWD, the main outcome in Schlenk et al. (2011), Messier et al. (2013), Miller et al. (2006), Messier et al. (2004), Gill et al. (2013,2015), Kuptniratsaikul, et al. (2019) is an parameter for aerobic capacity, not for walking/gait speed and this one is not included in the EWGSOP criteria. Please consider 6 min Walking test is not the same as walking (gait) speed measured on a 6m track.

So it is questionable how these studies would fit into your analyses?

Results:

-) The treatment groups received diet therapy alone, exercise alone, or their combination. One might consider dividing table 1 into these categories to make it a bit more visible to the reader. I am not sure if

-) Please consider recalculating 3.8 in line with the above mentioned remark.

-) 3.10: It was not mentioned that side effects and compliance will be reported. Please add to the methods section. Report how many studies explicitly mention/report side effects.

Discussion:

-)  I think the term “who have high sarcopenia risk” is probably not really suitable in the context of your MNA. Or did you assess this aspect?

-) please reconsider / rephrase the relevant paragraphs, if the walking speed and walking distance were erroneously combined.

Formatting:

-) Please delete ENTER in line 39 and try to format figures and tables accordingly. There are a lot of empty spaces. 

-) Figure 1. Please provide a meaningful figure legend, please harmonise typing style, additional records besides google scholar are not readable

-) Figure 3. I could not find figure 3 in the manuscript?

Author Response

[Nutrients]

Manuscript ID: nutrients-1200704

Author's Reply to the Review Report (Reviewer 1)

Comments and Suggestions for Authors

In the article “Relative efficacy of diet, exercise, and combined treatment for muscle mass and physical sarcopenia indices in adults with overweight or obesity and osteoarthritis: A network meta-analysis of randomized controlled trials” the authors aimed to (1) identify the relative effects of diet, exercise, and combined treatment on muscle mass, strength, and walking speed by using network meta-analysis (NMA) and (2) identify the optimal treatment by ranking the probability of each intervention type for people with overweight or obesity and knee or hip OA.

While the NMA might be of interest to a broader audience, I have severe concerns about the transparency of the publication strategies of the authors.

Major concerns:

The authors have published a previous study https://www.mdpi.com/2072-6643/12/8/2422/htm under an identical protocol number at Prospero and a similar study aim: “Thus, this study examined the effects of PS + ET on muscle mass and functional outcomes in older adults with OA in the lower extremity”.

In my opinion, this is of ethical relevance as the current study has not been registered at PROSPERO as stated in the article and the authors did not openly report differences/similarities between the studies. Therefore, this issue needs to be clarified before judging about its suitability for publication.  

Response

Thank you for the major comment regarding an ethical issue for our manuscript. We have submitted this study protocol at Prospero (protocol number CRD42021198023). In the revised manuscript, we corrected relative statements in the Methods section as follows:

Page 2, Lines 83–84.

“The study protocol was registered at PROSPERO (registration number: CRD42021198023).”

Please find below my suggestions for the manuscript, once the above issue has been adequately addressed.

Title:

I am not sure whether diet intervention is best suited in the context of your work. Did you address weight loss strategies, weight loss interventions, weight management …? This could be a bit more suitable (also to discriminate from protein interventions) and would fit at least to the information in your main table 1.

Response

Following with the constructive comment, we revised the title as follows:

“Relative Efficacy of Weight Management, Exercise, and Combined Treatment for Muscle Mass and Physical Sarcopenia Indices in Adults with Overweight or Obesity and Osteoarthritis: A Network Meta-Analysis of Randomized Controlled Trials”

I do not have any suggestions for the introduction, which is easy to read and follows a logical argumentation.

Methods – there are some linguistic inaccuracies, which have to be addressed in more detail:

-) Please provide the data bases also in the main text, i.e. Google scholar is mentioned in Figure 1, but not in the supplementary table S1, where various data bases are described.

Response

Following the reviewer’s comment, we added the searching terms of ClinicalKey database and Google scholar in the supplementary table S1. In addition, Figure 1 was revised accordingly.

-) You mentioned the differences between short/medium/ long-term therapies. Did you exclude acute interventions? Were there any restrictions according to age of participants? What’s about comorbidities? Please clarify in the inclusion/exclusion criteria.

Response

In this network meta-analysis, the studies which employed an acute intervention were included. We added the related statement and inclusion/exclusion criteria in section 2.3 as follows:

Page 3, Lines 106–110.

“(2) the study enrolled participants who aged ≥ 40 years, had body mass index (BMI) ≥ 25 kg/m2 and had a symptom or radiographic diagnosis of primary hip or knee OA. Participants were excluded if they had comorbidities such as rheumatic arthritis, neurological diseases (e.q., stroke, spinal stenosis), and substantial abnormalities in hematological, hepatic, or renal functions;”

Page 3, Lines 119–121.

“(7) the study conducted an acute intervention with a short period ranging from few days to 12 weeks, a medium-term or a long-term intervention with a treatment period of ≥ 6 months;”

-) Who decided about inclusion/exclusion of a study? Was it similar to the data extraction process?

Response

We added statements to clarify the process of study selection as follows:

Page 3, Lines 126–131.

“Study selection was initially performed by two authors (CDL and HCC) who independently screened and identified potentially relevant articles based on title and abstract. The full texts of all potentially eligible articles were examined to ensure they matched the inclusion criteria. In cases of inconclusiveness, the disagreements were resolved by discussions until consensus was obtained. A third author (THL) was consulted to discuss eligibility if the disagreement persisted.”

-) If LBM or FFM was not available, BF% was used to estimate the percentage FFM? You would need body mass/weight as well.

Response

We revised statements as follows:

Page 3, Lines 139–140.

“If LBM or FFM was not available, BF% and body weight was used to estimate the per-centage FFM.”

-) strength measures: knee extensors, knee flexors, and hip abductors are muscle groups not strength measures. What’s about power, peak torque, isometric/concentric/eccentric? What’s about upper extremities (ie. Bench press, which was the main outcome in Magrans-Courtney et al. (2011)).

Response

We revised statements as follows:

Page 3, Lines 141–144.

“Other strength measures were extracted on the basis of the following sequence of preference: concentric/eccentric power and peak torque, and maximum voluntary isometric contraction of knee extensors, knee flexors, and hip abductors; bench press and hand grip strength of upper extremity.”

-) One point is maybe the most critical issue here: 6MWD, the main outcome in Schlenk et al. (2011), Messier et al. (2013), Miller et al. (2006), Messier et al. (2004), Gill et al. (2013,2015), Kuptniratsaikul, et al. (2019) is an parameter for aerobic capacity, not for walking/gait speed and this one is not included in the EWGSOP criteria. Please consider 6 min Walking test is not the same as walking (gait) speed measured on a 6m track.

So it is questionable how these studies would fit into your analyses?

Response

Followed the reviewer’s comment, we removed the measure of 6 min walking distance from analyses of walking speed. We reperformed all analyses for walking speed outcome. Accordingly, all results regarding walking speed were revised and the statements were marked as red words in the Results section. Table 1, Figures 2, 5, and supplementary tables and figures were revised as well.           

Results:

-) The treatment groups received diet therapy alone, exercise alone, or their combination. One might consider dividing table 1 into these categories to make it a bit more visible to the reader.

Response

Table 1 is revised based on the reviewer’s comment.

I am not sure if

-) Please consider recalculating 3.8 in line with the above mentioned remark.

Response

Followed the reviewer’s comment, we removed the measure of 6 min walking distance from analyses of walking speed. We reperformed all analyses for walking speed outcome. Accordingly, all results regarding walking speed were revised and the statements were marked as red words in the Results section. Table 1, Figures 2, 5, and supplementary tables and figures were revised as well.

-) 3.10: It was not mentioned that side effects and compliance will be reported. Please add to the methods section. Report how many studies explicitly mention/report side effects.

Response

According to the reviewer’s comment, we added statements regarding side effects and compliance in Methods and Results sections as follows:

Page 4, Lines 161–163.

“We also examined the compliance for interventions as well as adverse events reported by the included studies.”

Page 18, Lines 457–464.

“Nonserious adverse events related to exercise intervention were observed by nine RCTs [59, 62, 65, 66, 71, 74, 78, 82, 90], among which most commonly conditions were train-ing-induced knee pain or muscle soreness of short duration (Tables S7). Three RCTs [60, 61, 78] reported adverse events related to the diet therapy which included food intol-erances and mild gastrointestinal reactions (Tables S7). In addition, among the 16 RCTs [59-62, 64-66, 71, 74-78, 82, 90, 91] which had information about adverse events, seven [59, 62, 74-76, 82, 91] and five [64, 75-77, 91] reported no any adverse event occurred during (or after) diet therapy and exercise intervention, respectively.”

Discussion:

-)  I think the term “who have high sarcopenia risk” is probably not really suitable in the context of your MNA. Or did you assess this aspect?

Response

According to the reviewer’s comment, we removed the words “who have high sarcopenia risk” and the statement was revised as follows:

Page 19, Lines 479–481.

“The primary goal of this study was to identify the relative efficacy of different diet therapies, exercise interventions, and combined treatments for muscle mass, strength, and functional outcomes in individuals with obesity and OA.”

-) please reconsider / rephrase the relevant paragraphs, if the walking speed and walking distance were erroneously combined.

Response

We removed the measure of 6 min walking distance from analyses of walking speed. Accordingly, the related statements in Discussion were revised as follows:

Page 19, Lines 486–489.

“(3) based on the cumulative ranking results, RMMR plus RET, RET alone, and AET alone were the optimal treatment strategy for muscle mass gain, strength gain, and walking speed recovery, respectively.”

Page 20, Lines 542–562.

“In this NMA, a series of meta-regression analyses were performed to identify the factors affecting treatment efficacy, and we found no significant moderation effects of age, BMI, gender, sample size, and methodological level (i.e., PEDro score). However, substantial moderating effects of some factors were noted. First, substantially but in-significantly minor treatment effects were exerted on muscle strength and walking speed in participants with an older age. In addition, a similar association was observed between age and muscle mass gain, although the finding was nonsignificant (Table S6). Previous studies have indicated that age-related changes in muscle mass are associated with a decline in strength and walking speed [104, 105]. Conversely, intervention-induced increases in muscle mass may improve the strength and walking ability of older people who have sarcopenia and frailty risks [106]. The previous results may explain the parallel associations between age and muscle mass change and between age and strength and walking speed that were discovered in this NMA. Second, compared with male sex, female sex may be associated with poorer walking speed outcomes after interventions in such patients with obesity and OA. Results in this NMA was in line with a previous systemic review which found such a sex-specific difference in physical mobility after a diet plus exercise intervention in older individuals with OA [107]. Our findings further indicate that sex mediates the relative treatment efficacy among different diet therapies, exercise interventions, and combined treatments for physical mobility, which may be explained by the sex-specific muscle adaptations in response to diet and exercise interventions for the older population [108-110] and those with OA [24, 111].”

Formatting:

-) Please delete ENTER in line 39 and try to format figures and tables accordingly. There are a lot of empty spaces. 

Response

Empty spaces were corrected in line 39 and all figures and tables were formatted as well.

-) Figure 1. Please provide a meaningful figure legend, please harmonise typing style, additional records besides google scholar are not readable

Response

Figure 1 was corrected and revised according to above comments.

-) Figure 3. I could not find figure 3 in the manuscript?

Response

We added Figure 3 in Page 14.

Reviewer 2 Report

This is a well-deigned metanalysis on the efficacy of diet, exercise, and combined treatment for muscle mass and physical sarcopenia indices in adults with overweight or obesity and osteoarthritis. The authors have been focused to an interesting topic in literature and have managed to avoid many of the pitfalls and bias in this approach. The methodology  and the presentation of the results are appropriate.

My concern is that the authors should clearly highlight the weaknesses of the previous studies and  recommend how to improve the design of future trials.

Also, I think that the discussion should be strengthened

English language is very good.

Author Response

[Nutrients]

Manuscript ID: nutrients-1200704

Author's Reply to the Review Report (Reviewer 2)

Comments and Suggestions for Authors

This is a well-deigned metanalysis on the efficacy of diet, exercise, and combined treatment for muscle mass and physical sarcopenia indices in adults with overweight or obesity and osteoarthritis. The authors have been focused to an interesting topic in literature and have managed to avoid many of the pitfalls and bias in this approach. The methodology and the presentation of the results are appropriate.

My concern is that the authors should clearly highlight the weaknesses of the previous studies and recommend how to improve the design of future trials.

Also, I think that the discussion should be strengthened

English language is very good.

Response

Thank you for the comprehensive review and constructive comments regarding our manuscript. We have added statements in Discussion section as follow:

Page 20, Lines 490–504.

“The present NMA can be considered clinically useful specifically because of the vast number of available treatment strategies and compositions for overweight or obese people who have lower-extremity OA. In the conservative pairwise me-ta-analysis coupling with multiple independent head-to-head trials, it is difficult to determine which treatment is the most efficacious [97]. By contrast, NMA provides consistent estimates of the relative treatment effects compared with each other using both direct (i.e., conservative pairwise meta-analysis) and indirect evidence without double counting the participants [98]. In addition, with a specific treatment such as RMMR plus MET, the inclusion of active comparators (i.e., RMMR alone or MET alone) is important as it encourages optimal clinical decision among different treatment op-tions by clinicians for overweight or obese patients with OA. Even in the absence of head-to-head RCTs, this NMA provides clinicians with the bottom-line knowledge re-garding the best available evidence on the comparative efficacy among different diet, exercise, and combined treatments for overweight or obese people with OA, especially those who are undergoing weight management.”

Round 2

Reviewer 1 Report

Dear authors, 

thank you for revising the manuscript and addressing the respective ethical and scientific concerns. 

I am just wondering about the numbers of included studies. Obviously, the authors conducted a new (hand) search as some studies containing only 6MWD (i.e. Messier et al., 2004) have been excluded. Nevertheless, the number of included studies remained the same with n = 34. 

-) Please confirm the final number of studies in the main body of the manuscript as well as in the abstract and the PRISMA flow chart. Why did the final number of studies stick to 34 if obviously some studies had to be excluded after the revision? Please add the date of the final search to enhance reproducibility of the paper. 

-) Please delete/adapt the hint to walking distance - In my opinion this does not make sense with respect to gait speed (lines 146/147). 

Thanks for considering these issues.

Kind regards 

Author Response

[Nutrients]

Manuscript ID: nutrients-1200704

Author's Reply to the Review Report (Reviewer 1)

Comments and Suggestions for Authors

Dear authors, 

thank you for revising the manuscript and addressing the respective ethical and scientific concerns. 

I am just wondering about the numbers of included studies. Obviously, the authors conducted a new (hand) search as some studies containing only 6MWD (i.e. Messier et al., 2004) have been excluded. Nevertheless, the number of included studies remained the same with n = 34. 

-) Please confirm the final number of studies in the main body of the manuscript as well as in the abstract and the PRISMA flow chart. Why did the final number of studies stick to 34 if obviously some studies had to be excluded after the revision? Please add the date of the final search to enhance reproducibility of the paper. 

Response

Thank you for the comment. In the first-round revised manuscript, we excluded 2 studies containing only 6MWD (i.e., Messier et al., 2004; Schlenk et al., 2011). In addition, we performed an updated search (on May 12, 2021) and added two studies which enrolled overweight patients with knee or hip osteoarthritis (Reference 85: Tak et al., 2005; Reference 90: Wallis et al., 2017) and reported walking speed outcome. Therefore, the final sample of this network meta-analysis was 34 trials. The PRISMA flow chart (Figure 1) as well as the analyzed results has been revised accordingly. Finally, we added the statement regarding date of the updated final search in Methods section 2.1 (Page 2, line 86 in main text) and Table S1.

-) Please delete/adapt the hint to walking distance - In my opinion this does not make sense with respect to gait speed (lines 146/147). 

Response

We deleted the term “walking distance” as follows:

Page 3, Lines 145–146.

“Walking speed was assessed using gait and walking parameters (e.g., walking time).”
